# Participation of Amyloid and Tau Protein in Post-Ischemic Neurodegeneration of the Hippocampus of a Nature Identical to Alzheimer’s Disease

**DOI:** 10.3390/ijms22052460

**Published:** 2021-02-28

**Authors:** Ryszard Pluta, Liang Ouyang, Sławomir Januszewski, Yang Li, Stanisław J. Czuczwar

**Affiliations:** 1Laboratory of Ischemic and Neurodegenerative Brain Research, Mossakowski Medical Research Institute, Polish Academy of Sciences, 02-106 Warsaw, Poland; sjanuszewski@imdik.pan.pl; 2State Key Laboratory of Biotherapy and Cancer Center, West China Hospital, and Collaborative Innovation Center of Biotherapy, Sichuan University, Chengdu 610041, China; ouyangliang@scu.du.cn (L.O.); yangli@stu.scu.edu.cn (Y.L.); 3Department of Pathophysiology, Medical University of Lublin, 20-090 Lublin, Poland; stanislawczuczwar@umlub.pl

**Keywords:** brain ischemia, hippocampus, amyloid, tau protein, α-synuclein, secretases, presenilin, neuronal death, neurodegeneration, amyloid plaques, neurofibrillary tangles, dementia, genes

## Abstract

Recent evidence suggests that amyloid and tau protein are of vital importance in post-ischemic death of CA1 pyramidal neurons of the hippocampus. In this review, we summarize protein alterations associated with Alzheimer’s disease and their gene expression (*amyloid protein precursor* and *tau protein*) after cerebral ischemia, as well as their roles in post-ischemic hippocampus neurodegeneration. In recent years, multiple studies aimed to elucidate the post-ischemic processes in the development of hippocampus neurodegeneration. Their findings have revealed the dysregulation of genes for *amyloid protein precursor*, *β-secretase*, *presenilin 1* and *2*, *tau protein*, *autophagy*, *mitophagy,* and *apoptosis* identical in nature to Alzheimer’s disease. Herein, we present the latest data showing that amyloid and tau protein associated with Alzheimer’s disease and their genes play a key role in post-ischemic neurodegeneration of the hippocampus with subsequent development of dementia. Therefore, understanding the underlying process for the development of post-ischemic CA1 area neurodegeneration in the hippocampus in conjunction with Alzheimer’s disease-related proteins and genes will provide the most important therapeutic development goals to date.

## 1. Introduction

Hippocampus in animals and humans is one of the brain’s most sensitive areas to ischemia, especially for pyramidal neurons in the CA1 area [1,2,3,4]. Pyramidal neurons in the CA1 area of the hippocampus disappear selectively 2 to 7 days following global brain ischemia [1,2,5]. After middle cerebral artery occlusion, neuronal damage and loss can also be observed in the CA1 area of the hippocampus [6,7]. Currently, the processes underlying this phenomenon have not been fully elucidated. Researchers have found that the CA1 region is characterized by a low capillary density compared to the CA3 region of the hippocampus [8]. Meanwhile, numerous studies have demonstrated other factors involved in neuronal death after ischemia in the CA1 region of the hippocampus, including changes in calcium levels, glutamate-mediated excitotoxicity, oxidative stress caused by excess production of reactive oxygen species, and neuroinflammatory response [9,10,11,12,13]. It is well known that the death of neurons in the CA1 area is accompanied by gliosis, which is characterized by the reaction of astrocytes and microglia [10,13,14,15]. Astrocytes, the most abundant glial cells in the brain, support neuronal functions such as K^+^ buffering, H^+^ control, neurotransmitter uptake, blood-brain barrier regulation, and brain water transport [16]. In the ischemic brain, astrocytes respond to ischemic damage by restoring homeostasis. On the other hand, they are involved in the production of pathogenic substances [17]. Selective astrocyte dysfunction has been confirmed to be associated with the death of neurons after ischemic injuries [18]. Despite the knowledge of the above mechanisms, cerebral ischemia is a debilitating and progressive injury as it damages the CA1 area of the hippocampus, which is an important area of learning and memory. Survivors of cerebral ischemia suffer from irreversible damage to the CA1 area, eventually resulting in learning and memory deficits [19,20]. So far, there is no specific treatment option that can prevent the subsequent neurofunctional dysfunction due to the damage of the hippocampal CA1 region in particular. Until recently, based on the above-mentioned mechanisms, many researchers are still trying to develop effective neuroprotective agents against experimental cerebral ischemia, with more attention paid to changes in the hippocampus [21,22]. The latest findings show that most neuroprotective agents that have been proved to be effective in experimental models of cerebral ischemia do not provide a significant benefit in clinical trials [23].

Therefore, further exploration of irreversible neuronal death in the CA1 area of the hippocampus following ischemia and new therapies for prevention is urgently needed. Emerging evidence has suggested that abnormal protein folding and aggregation may be key irreversible neuropathological events increasing the neurotoxicity of neurons after ischemia [24,25,26,27,28]. Recently, one study has indicated that focal cerebral ischemia in rats induces protein aggregation in ischemic neurons after 1 h of reperfusion, which may last until necrotic neuronal death 24 h after reperfusion [29]. In addition, the abnormal accumulation of ribosomal protein continues until the onset of delayed neuronal death, 2 days after ischemia [30]. It should be assumed that the deposition and accumulation of unwanted protein aggregates may eventually contribute to the death of neurons following ischemia [24,25,26,27]. Increasingly more data suggest that brain ischemia with recirculation may trigger the pathology of the folding protein characteristic of Alzheimer’s disease through the production and accumulation of amyloid and tau protein [31,32,33]. Meanwhile, the findings present changes in the expression of genes involved in the amyloidogenic metabolism of the amyloid protein precursor, as well as changes in the tau protein and its gene in the CA1 area of the hippocampus after ischemia [32,34]. In addition, we pay attention to whether amyloid and tau protein participate in the death of pyramidal neurons in the CA1 region of the hippocampus after ischemia and dementia development. In this paper, in light of the new data, we will focus on the behavior of amyloid and tau protein and their genes in the hippocampal CA1 area after brain ischemia with reperfusion. Proteins must be properly structured to perform their physiological functions. Thus, aggregated proteins usually have little or no biological activity, but exert negative impacts on individual health and safety. Unfortunately, compared with neurodegenerative diseases, the mechanisms of protein aggregation after cerebral ischemia and methods for prevention remain unclear. Moreover, the influence of neurotoxicity of protein aggregates following cerebral ischemia is not well understood. Here, we review the latest knowledge on proteins folding and aggregation in the CA1 area of hippocampus (amyloid, tau protein, and α-synuclein) and their internal role in the post-ischemic brain. All our findings may lay a solid foundation for future research. 

## 2. Amyloid Protein Precursor, β-Secretase, Presenilin 1, Presenilin 2, and Tau Protein Genes

In the CA1 area of the hippocampus, the expression of *amyloid protein precursor* gene is lower than that of the control 2 days after ischemia (Table 1) [32]. In contrast, 7–30 days following post-ischemic brain injury, the expression of the above gene is higher than the control values (Table 1) [32]. Two days after ischemia, the expression of the *amyloid protein precursor* gene decreased to a minimum of −0.542-fold change, and on day 7 and 30, the expression increased to a maximum of 0.746 and 0.623-fold change, respectively [32]. The expression of *β-secretase* is higher than the control level at 2–7 days after ischemia in the CA1 area of the hippocampus (Table 1) [32]. The expression of *β-secretase* 30 days after ischemia is lower than the control values (Table 1) [32]. Two and 7 days after ischemia, the *β-secretase* gene expression increased to a maximum of 3.916 and 0.952-fold change, respectively, but after 30 days the expression had dropped to a minimum of −0.731-fold change [32]. In the CA1 region, the expression of *presenilin 1* and *2* genes increases significantly at 2–7 days after the ischemic episode (Table 1) [32]. Moreover, 30 days after ischemic injury, the expression of *presenilin 1* and *2* is lower than the control values (Table 1) [32]. Two and 7 days after ischemia, the *presenilin 1* gene expression increased to a maximum of 2.203 and 0.874-fold change, respectively, but at day 30 the expression had dropped to a minimum of −1.093-fold change [32]. Two and 7 days after ischemia, the *presenilin 2* gene expression increased to a maximum of 3.208 and 0.709-fold change, respectively, but at day 30 the expression had dropped to a minimum of −0.728-fold change [32]. In neurons of the presented region, the expression of *tau protein* gene is higher than the control values 2 days following ischemic brain injury (Table 1) [34]. At 7–30 days following ischemia, the expression of *tau protein* gene decreases obviously compared with the control values (Table 1) [34]. The expression of the *tau protein* gene was elevated to a maximum of 3.297-fold change on the second day after cerebral ischemia, but a minimal expression was noted at −0.492 and −0.351-fold change, respectively, on day 7 and 30 [34].

Alterations in the mean expression level of *amyloid protein precursor* gene are statistically significant between 2 and 7, between 2 and 30, and between 7 and 30 days after hippocampal ischemia [32]. Changes in the mean level of *β-secretase* gene expression show statistically significant differences between 2 and 7, between 2 and 30, and between 7 and 30 days post-ischemia [32]. Alterations in the mean expression level of *presenilin 1* gene are statistically significant between 2 and 30 and between 7 and 30 days following ischemia [32]. Changes in the mean expression level of *presenilin 2* gene are statistically significant between 2 and 7, between 2 and 30, and between 7 and 30 days after ischemia [32]. Significant correlations between the expression of genes in the CA1 area of rat hippocampus at 2, 7, and 30 days after ischemia are observed for *presenilin 1* and *presenilin 2*, *presenilin 1* and *β-secretase,* as well as *presenilin 2* and *β-secretase* [32]. The statistical significance of changes in the expression of *tau protein* gene after the CA1 area ischemia is between 2 and 7 and between 2 and 30 days [34].

## 3. Amyloid Staining

After experimental post-ischemic brain injury, with a survival of up to 0.5 years, intra- and extracellular staining in the CA1 area is present for both C- and N-terminal of the amyloid protein precursor and amyloid [1,31,35,36,37,38,39,40,41]. Staining of different fragments of the amyloid protein precursor has been observed in CA1 neurons, astrocytes, microglia, and oligodendrocytes [1,38,40,42,43,44]. From 0.5 to 1 year following ischemia, only C-terminal staining of the amyloid protein precursor and amyloid can be observed [1,2,39,45]. Therefore, it is believed that astrocytes with a huge accumulation of different fragments of the amyloid protein precursor are involved in the development of the glial scar [1,40,44]. In addition, reactive astrocytes with abnormal amyloid deposits may also participate in the pathological repair of brain tissue following ischemia, accompanied by the death of astrocytes [1,31,40,46]. Extracellular deposits of the different sites of the amyloid protein precursor appear to be small dots or characteristic diffuse amyloid plaques [1,2,31,39,40]. Diffuse amyloid plaques in response to experimental ischemia in the CA1 area of the hippocampus are not transient due to the reason that diffuse amyloid plaques have been shown to transform into senile plaques approximately 1 year after the ischemic brain injury [47]. The accumulation of amyloid in neurons and additionally in astrocytes indicates the pathological metabolism of the amyloid protein precursor in neurodegeneration after hippocampal ischemia [31,37,44,48,49]. These data clearly suggest that the gradual post-ischemic amyloid deposition in the hippocampus may be responsible for secondary neurodegenerative mechanisms. Meanwhile, progressive neuronal death may further worsen the prognosis after ischemia [2,38,39,45,50,51]. Currently, it has been confirmed that following brain ischemia, amyloid is formed as a result of damage and death to pyramidal neurons of the hippocampus [36]. Its neurotoxic effects likely contribute to the development of Alzheimer’s disease-type dementia. Amyloid is a neurotoxic substance. After local ischemia, it initiates pathological intracellular mechanisms in neurons, astrocytes, oligodendrocytes, and microglia, leading to the death of neurons and neuroglial cells [52].

The accumulation of amyloid has also been noted in post-ischemic human brains [53,54,55,56]. Diffuse and senile amyloid plaques are predominant in hippocampus [4,53,54,55,56]. According to another post-ischemic study of the human brain, staining for various amyloids has been found in the ischemic hippocampus [56]. In post-ischemic patients, enhanced staining for β-amyloid peptide 1–40 and 1–42 is observed in the ischemic hippocampus [56]. Increased staining of various amyloids likely contributes to the progression of ischemic neurodegeneration and ultimately promotes the development of Alzheimer’s disease.

## 4. Tau Protein Staining

Strong staining of tau protein was found in hippocampal neurons after experimental ischemia-reperfusion [41,57]. Additionally, tau protein was widely stained in astrocytes, oligodendrocytes, and microglia after ischemia [58,59]. These results suggested that some pyramidal neurons show changes in the tau protein after hippocampal ischemia-reperfusion injury, indicating that these cells are in the primary neuropathological stage of the ischemic process. Another study found that the dysfunctional tau protein could suppress the transport of amyloid from the neurons body through axons and dendrites, resulting in amyloid deposition in the neurons body [60]. Existing studies have shown that the hyperphosphorylated tau protein plays a dominant role in neurons during ischemia, accompanied by neuronal apoptosis [58,59,61,62,63]. Thus, the apoptosis of pyramidal neurons after cerebral ischemia is directly related to the hyperphosphorylation of the tau protein. Furthermore, cerebral ischemia is related to the development of paired helical filaments [64], neurofibrillary tangle-like [61,62,63], and neurofibrillary tangles [65,66] after ischemia. After the hippocampal ischemic injury, neurons showed dysfunction of the tau protein and the development of neurofibrillary tangles [41,58,59,61,62,63,64,65,67,68,69]. The above data constitute the pathological basis for the development of dementia after cerebral ischemia-reperfusion with the Alzheimer’s disease phenotype.

## 5. Alpha-Synuclein Staining

Cerebral ischemia leads to the accumulation of α-synuclein around the blood vessels in the hippocampal CA1 region [70,71]. In addition, α-synuclein was strongly stained in the glial cells in the degenerative segment of the hippocampus after ischemia [70]. Thus, α-synuclein may be a necessary protein to induce neurodegeneration after ischemia [72]. In the Hashimoto and Masliah [73] study, the pathological accumulation of α-synuclein at presynaptic terminals causes disruption of the synaptic activity, resulting in cognitive impairment. The neuropathological interaction of α-synuclein may disrupt the synaptic function and may lead to inducing retrograde neuronal death after the ischemic brain injury [72].

## 6. Neuropathophysiology and Neuropathology

To date, the hippocampus has always been the preferred area to study the effects of ischemia for the following reasons. Firstly, the hippocampus, as a brain structure, exhibited similar post-ischemic neuropathological changes to those observed in Alzheimer’s disease. Secondly, as an important brain sector for human memory and learning, the hippocampus damage causes the same dementia in Alzheimer’s disease and after ischemia. Thirdly, the hippocampal region of CA1 is one of the brain regions prone to cerebral ischemia and Alzheimer’s disease. In vivo studies, cerebral ischemia could cause selective death of pyramidal neurons in the hippocampal CA1 region [1,2,48,49,74,75]. Additionally, the excessive release of excitatory amino acids and intracellular calcium overload has been observed after ischemic damage to the hippocampus [9,76,77,78]. This post-ischemic phenomenon, known as excitotoxicity, is caused by the excessive release of glutamate [9]. The release of glutamate from presynaptic terminals and its insufficient reuptake increased the concentration of glutamate in the extracellular space of the hippocampal CA1 region [9]. Therefore, the glutamate receptors were over-stimulated, which caused a huge influx of calcium ions to neurons through calcium channels [9,76,77,78]. An enzyme activated by intracellular calcium determines the survival and/or death of neurons. Furthermore, activation of related proteases (endonucleases, phospholipases, and nitric oxide synthase) results in damage to the nuclear membrane and other cytoplasmic organelles, which in turn leads to neuronal damage and/or death. Classically, the death of neurons in the CA1 region after acute ischemic injury is necrotic, and the next stage of neuronal death is caused by apoptosis. Mechanistically, necrosis is the result of energy loss and osmotic homeostasis, involving a huge number of pyramidal neurons in the CA1 region. After ischemia, the neurons swelled by absorbing too much water, which caused the rupture of cytoplasmic membranes and the outflow of neuronal contents to the extracellular space [79,80]. In contrast, the DNA cleavage in neurons is a late phenomenon in the process of serine protease-dependent necrosis [81]. The occurrence of necrosis is mainly due to the sharp decrease of neuronal energy and the decrease of blood glucose level after ischemia. The time after ischemia determines whether neurons die due to necrosis or apoptosis.

In the Nitatori et al. [82] study, apoptosis was observed in the CA1 region of hippocampal pyramidal neurons after 4 days of ischemia. There are two main processes that trigger apoptosis after neuronal ischemia: Receptor-mediated apoptosis and the mitochondria-related apoptosis pathway. In ischemia-sensitive nerve cells, cytochrome *c*, as a key factor in the mitochondrial apoptosis pathway is released into the cytoplasm [83,84]. Specifically, mitochondria participated in the process of necrosis or apoptosis according to the time after ischemia. Caspase-3 also played a key role in apoptotic neuronal death after ischemic brain damage [85,86,87,88]. Nevertheless, the relationship of autophagy and mitophagy with post-ischemia apoptosis of neurons should be taken into account [86,87,88]. Firstly, the receptor activated during ischemia stimulates procaspase-8, while caspase-8 stimulates caspase-3, and then caspase-3 cleaves poly (ADP-ribose) polymerase-1 and activates DNAase, which causes DNA fragmentation and death of neuronal cells [85,89]. In addition, another pathway of neuronal death called necroptosis was described after ischemia [90]. In this pathway, post-ischemic neuronal cells showed signs of necrotic and apoptotic phenomena in the same neuron at the same time [91]. Secondly, in the process of autophagy programmed neuronal death, autophagosomes and autolysosomes were observed in dead neurons [92]. It has been suggested that this pathway may work in two directions, namely that autophagy may protect neurons from death through apoptosis and act as a death trigger [93]. In the first option, through the lysosomal degradation of their own cytoplasmic organelles, neurons receive substrates both for energy generation and for the synthesis of necessary proteins. In the second case, it is responsible for the death of neurons programmed by autophagy [86,87,88,93,94]. Recent evidence suggests that mitophagy, along with caspase-3, plays a significant role in the ischemic death of pyramidal neurons in the hippocampal CA1 region (Table 2) [86,87,88,93,95]. The maximum expression of the *autophagy* gene in the hippocampal CA1 region 2 and 7 days after ischemia was 0.236 and 0.107-fold change, respectively (Table 2) [88]. Thirty days after ischemia, the gene expression was reduced to a minimum of −0.089-fold change (Table 2) [88]. There were no statistically significant changes between the levels of gene expression at different post-ischemic times [88]. The maximum expression of the *mitophagy* gene in the hippocampal CA1 region 2 days after ischemia was a 1.371-fold change (Table 2) [88]. Seven and 30 days after ischemia, the gene expression decreased to a minimum of −0.157 and −0.126-fold change, respectively (Table 2) [88]. Significant changes between the levels of gene expression were seen between 2 and 7 and between 2 and 30 days after ischemia [88]. The maximum expression of the *caspase-3* gene in the hippocampal CA1 area 2 and 7 days after ischemia was 4.417 and 0.145-fold change, respectively (Table 2) [88]. Thirty days after ischemia in the above area, the gene expression was reduced to a minimum −0.202-fold change (Table 2) [88]. Changes in the caspase-3 gene expression between 2 and 7, between 2 and 30, and between 7 and 30 days were statistically significant [88].

The death of pyramidal neurons in the CA1 subfield of the hippocampus occurs between 2 and 7 days after an episode of ischemia [74]. In vivo models, cerebral ischemia could cause the characteristic damage of pyramidal neurons in the hippocampal CA1 region [1,2,48,49,74,75]. Prolonging the survival time of rats after cerebral ischemia for up to 2 years will cause changes in nerve cells in the hippocampus to ischemic non-sensitive areas (such as CA2 and CA3 areas) [2]. After ischemia, the loss of neurons and the decrease of acetylcholine level were observed in the hippocampal CA1 subfield. These results suggested that neuronal death may also be related to the lack of neuronal stimulation and cholinergic transmission [9,96].

Strong reactions of astrocytes and microglia were observed in the hippocampal CA1 region after ischemia [1,10,13,48,49,97]. Moreover, the response of astrocytes in the hippocampal CA1 region to cytokines was enhanced [97]. These data indicated that the increase of neuroinflammatory mediators in astrocytes is directly related to the selective sensitivity of hippocampal pyramidal neurons to ischemia [97,98]. In the Touzani et al. [98] study, the expression of interleukin-1 receptor in neurons increased during ischemia. Therefore, neurons in the ischemic hippocampal sensitive region may be the target for the production of interleukin-1β by astrocytes [98]. Furthermore, in hippocampal ischemia and Alzheimer’s disease, interleukin-1 is the main factor that stimulates the metabolism of amyloid protein precursor in neurons and stimulates the release of neuroinflammatory mediators, which cause irreversible damage to the neural network. Specifically, neuroinflammatory mediators can activate a self-sustaining cycle, leading to neurodegeneration specific to Alzheimer’s disease [99]. Additionally, amyloid produced after ischemia promotes the release of neuroinflammatory mediators by microglia [2,31,100]. In the hippocampus, the activation of glial cells preceded neuronal injury/death, and lasted for a long time after the ischemic event [10,13].

In the Neumann et al. [101] study, observations by electron microscopy showed synaptic changes in the hippocampal CA1 region after ischemia. In addition, several studies also have demonstrated that ischemia stimulates synaptic autophagy (Table 2), which is associated with the death of neurons in the CA1 area of the hippocampus following ischemia [86,88,94,102,103]. Excitatory synaptic transmission was also decreased after ischemia in the hippocampal CA1 region [9]. The increase of intracellular calcium level after ischemia activated calpains in neurons, and calpain target proteins were present at glutamate and gamma-aminobutyric acid synapses. After cerebral ischemia, calpain broke down presynaptic and postsynaptic proteins in the brain. Therefore, the cleavage of calpain-related proteins leads to the death of neurons following ischemia [104].

Ischemia in the hippocampus increases the permeability of the blood-brain barrier, which may lead to blood diffusion and leakage through the necrotic wall [77,78,105,106,107,108,109,110,111,112,113,114,115,116,117,118]. Ischemic damage of the blood-brain barrier can cause toxic reactions triggered by extravasations of neurotoxic proteins such as amyloid, and damage caused by leakage of blood cells [119,120,121,122,123,124]. Specifically, in a state of ischemia, amyloid, which has the ability to cross the blood-brain barrier, is toxic to specific groups of neurons, such as the hippocampus, which in turn causes damaged nerve cells to produce more amyloid [36].

The response of the hippocampus to ischemia is a strong inflammatory response that lasts for a long time [10,13,125]. Immediately following ischemia, neurons and glial cells induced molecular responses, leading to the activation of astrocytes. Activated astrocytes replicated quickly, changing their shape and function [126]. These astrocytes secreted pro-inflammatory cytokines (e.g., interleukin-1β, tumor necrosis factor-α), metalloproteinase, and chemokines [127]. Astrocytes stimulated by a single cytokine, such as interleukin-1β, can produce an amyloid protein precursor, but amyloid production is regulated differently [128]. Some data indicate that astrocytes can produce large amounts of β-amyloid peptides 1–40 and 1–42 after the treatment with cytokines [128]. This suggests that astrocytes may be an important source of amyloid in the presence of certain combinations of inflammatory cytokines. This effect may be mediated by interferon-γ in combination with the tumor necrosis factor-α or interleukin-1β, this system appears to trigger amyloid production by promoting the β-secretase cleavage of the amyloid protein precursor [128]. Another study in transgenic mice showed the age-dependent upregulation of the amyloid protein precursor in astrocytes [129]. Moreover, astrocyte cell models have shown that the pretreatment with β-amyloid peptide 1–42 causes autonomic changes downstream of astrocytes, including the upregulation of amyloid protein precursor and β-secretase levels, as well as prolonged amyloidogenesis [129]. Substances released by astrocytes, especially matrix metalloproteinase and interleukin-1β, increased the destruction of the ischemic blood-brain barrier, thereby increasing the transfer of leukocytes from the blood to brain tissue [130,131]. The influx of these cells will further lead to progressive ischemic damage in the hippocampus [10,13]. Microglia, similar to astrocytes, belongs to the first line of defense and is activated immediately after ischemia [125]. After cerebral ischemia, the activation of hippocampal CA1 neuroglial cells can last up to 2 years [13,132]. The activated microglia changed their shape into amoeba and gain phagocytosis [125]. Microglia secreted pro-inflammatory substances to increase the permeability of the blood-brain barrier, including tumor necrosis factor-α, matrix metalloproteinase-9, and interleukin-1.

Several evidences indicated that transient hippocampal ischemia in humans and animals leads to a widespread loss of CA1 neurons that are selectively sensitive to ischemia [1,2,3,4,133]. Ischemic changes in the hippocampus refer to the process that progresses with the increase of reperfusion time following an ischemic episode [2,13,133]. The degree of hippocampal atrophy is proportional to the duration of cerebral ischemia and reperfusion [1,2,3,4,45,133,134]. These phenomena explain the complete disappearance of pyramidal neurons in the CA1 region and are accompanied by an increase in the permeability of the blood-brain barrier in the early and late stages of ischemia [107,114,115,135].

## 7. Dementia

Neurodegeneration of the CA1 area of the hippocampus in animals due to ischemia with recirculation causes dementia [50,136,137,138,139]. In vivo models of cerebral ischemia, a spontaneous restoration of sensorimotor functions is observed, but the process of dementia is irreversible [50,140,141]. In the early stage, locomotor hyperactivity has been documented in animals after hippocampal ischemia [142,143]. Hyperactivity is an indicator of hippocampal pyramidal neuronal death after ischemia [142,144]. Prolonged ischemia or repeated ischemia could lead to a persistent hyperactivity and habit disorder, which is directly related to the increase in the number of neuronal deaths in the hippocampus and the progressive development of inflammation in the hippocampus [10,13,135,141,145]. In addition, hippocampal ischemia could result in the loss of reference memory, spatial memory, and working memory, as well as the permanent impairment of learning and memory abilities [50,145]. Additionally, there is a positive proportional relationship between the progress of cognitive impairment and the time after ischemia [50,144,146]. Dementia was associated with a complete atrophy of the hippocampus, especially in its CA1 region, accompanied by a progressive loss of massive number of neurons in the CA3 region [1,2,45,48,49,51,134,147,148], perirhinal cortex, and amygdala [136].

In clinical research, dementia caused by an ischemic stroke is accompanied by hippocampal neuronal atrophy [3]. The volume of pyramidal neurons in the CA1 and CA2 regions of the hippocampus in patients with post-stroke dementia is 10 to 20% lower than that in the elderly with normal cognition [3]. In addition, the volume of neurons in the delayed dementia group after stroke was 20% smaller than that in the group without post-stroke dementia [3]. The volume of neurons in hippocampal CA1 and CA2 regions of ischemic stroke patients with delayed dementia was significantly smaller than that of stroke survivors with normal cognition, learning, and memory. These results suggest that the reduction in the volume of neurons in the hippocampus reflects characteristic changes in post-stroke patients that accelerate cognitive decline. Furthermore, this was confirmed by a positive correlation between the volume of neurons in stroke survivors and the results of cognitive tests [3]. It is speculated that the decrease in the volume of neurons, especially in the CA1 region, reflects the pathological process of the hippocampus that leads to cognitive impairment. Nevertheless, in the hippocampal subregion, amyloid deposition is much more abundant in the CA1 region than in the CA2 and CA3 regions [4].

## 8. Conclusions

The paper presents the genotypic and phenotypic neurodegeneration of Alzheimer’s disease type in the hippocampal CA1 region after cerebral ischemia, such as neuropathology, neuropathophysiology, amyloid, tau protein, and its genes, which play a key role in the occurrence and development of dementia (Figure 1). It shows changes in proteins and their genes expression, i.e., the *amyloid protein precursor*, *β-secretase*, *presenilin 1*, *presenilin 2,* and *tau protein* in rats following ischemia in the CA1 region of the hippocampus, as well as genes involved in the death of neurons after ischemia in the hippocampus, i.e., *autophagy*, *mitophagy,* and *apoptosis*. Data show that ischemia causes the death of pyramidal neurons in the hippocampus in an amyloid-dependent manner (Figure 1). These changes are associated with amyloid deposition in the intra- and extracellular space and a massive loss of neurons followed by hippocampus atrophy, eventually leading to dementia of the Alzheimer’s disease type (Figure 1). Following ischemia with reperfusion, the formation of amyloid plaques in the hippocampus is very likely due to an increased production and plasma influx, and a decreased clearance of amyloid from the brain. The above data suggest a direct relationship between ischemia and increased levels of amyloid in the hippocampus, such as diffuse and senile amyloid plaques [4,31,47]. Based on the presented data, it can be concluded that the ischemic damage to the hippocampus affects the processing of the amyloid protein precursor both at the gene and protein levels, leading to the accumulation of neurotoxic amyloid plaques [4,31,47]. Moreover, the data revealed that the CA1 ischemia altered the phosphorylation level of tau protein and its gene expression. It should be emphasized that the deregulated tau protein is also involved in the death of neurons in the hippocampus (Figure 1). Moreover, the existing evidence documents the formation of neurofibrillary tangles after ischemia (Figure 1). The above data are supported by the observed elevated cyclin-dependent kinase 5 levels following the ischemia-reperfusion brain injury. This suggests a relationship between tau protein dysfunction and the onset of pyramidal neuronal death in the hippocampus after ischemia (Figure 1). The above data indicate the regulation of ischemic death of pyramidal neurons in the hippocampus in a stage dependent manner of the tau protein (Figure 1). Moreover, new data suggest that the development of post-ischemic tauopathy may be one of the major mechanisms of cognitive impairment in ischemic brain neurodegeneration [149]. This study suggests that the development of uncontrolled tauopathy in the brain after ischemia may be associated with increased tau protein hyperphosphorylation not only due to an ischemic episode, but also due to an impaired clearance of tau protein oligomers from the brain tissue [149].

The conclusions drawn from the study of ischemia-induced Alzheimer’s disease-associated neuropathology, proteins, and their genes in the hippocampus, which contribute to the injury and death of pyramidal neurons, as well as the generation of the amyloid plaques and neurofibrillary tangles development, are vital for the development of new therapeutic targets in the therapy of neurodegenerative diseases. Since amyloid deposits and dysfunctional tau protein are also important in the pathology of the hippocampus in Alzheimer’s disease patients [4], further research in this field will be very useful. Animal models of brain ischemia with a damage to the hippocampus appear to be a recommended experimental approach in determining the role of aggregating and folding proteins and their genes in the neurodegenerative disease. Evidences have suggested that targeting amyloid deposition and dysfunctional tau protein can present beneficial effects in the treatment of Alzheimer’s disease. However, the development and discovery of new small molecular therapeutic chemotypes and analogues for brain ischemia and Alzheimer’s disease is still challenging.

The influence of amyloid and tau protein in post-ischemic hippocampus neurodegeneration is a clear fact, but from the available publications much remains to be clarified regarding the final relationship between hippocampal ischemia and Alzheimer’s disease. This review summarizes previous works suggesting that ischemia probably plays a significant role in triggering and/or accelerating the Alzheimer’s disease type cognitive decline. On these grounds, a better understanding of the molecular mechanisms of the progression of injury after hippocampal ischemia is still needed. Moreover, obtaining more complete genetic, molecular, and cognitive profiles of patients is suggested to allow for future studies on an individual basis. In conclusion, the behavior of Alzheimer’s disease connected proteins and their genes in post-ischemic hippocampal neurodegeneration in patients should be investigated in the near future, as the lack of such data is a significant limitation of our current knowledge.

## Figures and Tables

**Figure 1 ijms-22-02460-f001:**
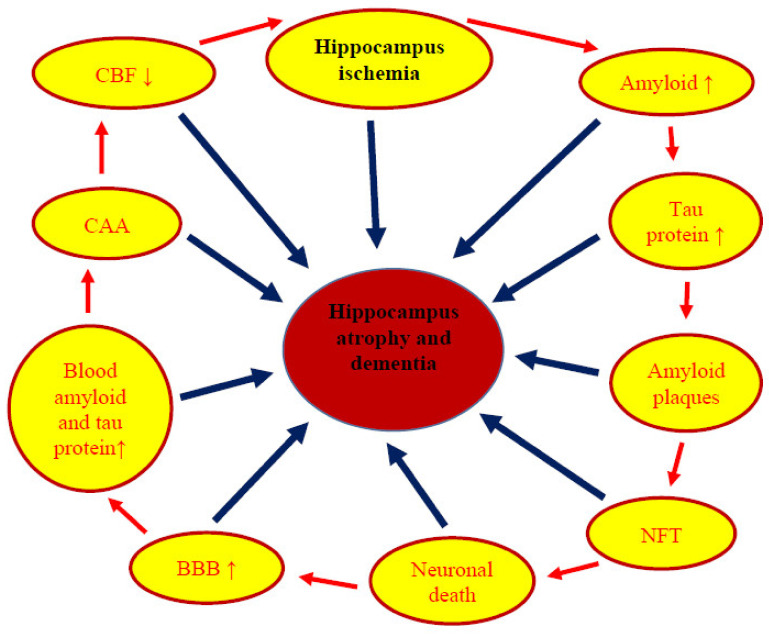
Pathological phenomena after ischemia in the hippocampus. ↑—Increase; ↓—decrease; NFT: Neurofibrillary tangles; BBB: Blood-brain barrier; CAA: Cerebral amyloid angiopathy; CBF: Cerebral blood flow.

**Table 1 ijms-22-02460-t001:** Changes in the expression of *amyloid protein precursor* (*APP*), *β-secretase* (*BACE1*), *presenilin 1* (*PSEN1*), *presenilin 2* (*PSEN2*), and *tau protein* (*MAPT*) genes in the rat CA1 area after brain ischemia.

	Genes	* APP *	* BACE1 *	* PSEN1 *	* PSEN2 *	* MAPT *
Survival						
**2 days**		↓	↑↑	↑	↑↑	↑↑
**7 days**		↑	↑	↑	↑	↓
**30 days**		↑	↓	↓	↓	↓

Expression: ↑ Increase; ↓ decrease.

**Table 2 ijms-22-02460-t002:** Changes in the expression of *autophagy* (*BECN1)*, *mitophagy* (*BNIP3*), and *apoptosis* (*CASP3*) genes in the CA1 area of hippocampus at different times after experimental brain ischemia.

	Genes	* BECN1 *	* BNIP3 *	* CASP3 *
Survival				
**2 days**		↔	↑	↑↑↑
**7 days**		↔	↔	↑
**30 days**		* ↔ *	↔	↓

Expression: ↑ Increase; ↓ decrease; ↔ oscillation around control values.

## Data Availability

Not applicable.

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
