# Peer review of "Participation of Amyloid and Tau Protein in Post-Ischemic Neurodegeneration of the Hippocampus of a Nature Identical to Alzheimer's Disease"

_ijms, 2021, doi:10.3390/ijms22052460_

Round 1
Reviewer 1 Report
The theme the authors are focusing on this manuscript is very interesting. The review focused with best clarity/cohesion. I liked that the authors analyze not only the total expression of proteins, but also consider their cellular and intracellular lacalization in the hippocampus.
Figure 1 spoils the impression of the review. Arrows cover the text. Figure 1 was done in a hurry.
Author Response
Review 1. -All text changes are in red and citation changes are highlighted in green. 1).The theme the authors are focusing on this manuscript is very interesting. The review focused with best clarity/cohesion. I liked that the authors analyze not only the total expression of proteins, but also consider their cellular and intracellular lacalization in the hippocampus. Ad.1.Done. 2). Figure 1 spoils the impression of the review. Arrows cover the text. Figure 1 was done in a hurry. Ad.2. Problems with figure 1 arose when the editors of the journal decided to change the templete. Currently, the figure is modified due to changes in the text of the work. We hope that the figure will not change during the editorial activities.
Reviewer 2 Report
This review addresses a relevant scientific issue, which is clearly presented. The authors have a long track of publications on this topic in good standing specialized journals. Although this review is well written, I have several concerns that should be addressed by the authors before I can recommend publication of this manuscript. My main concern is that the authors have recently published closely related reviews, which are cited as references in this work. Therefore, the authors should revise this manuscript to highlight the novel experimental facts and hypothesis that are not dealt with in their previous and recent reviews.
In addition, the analysis of the literature data looks very descriptive and the overall discussion is largely qualitative. This is a weakness in this work, since it is well known that ischemia-reperfusion injury is a complex and multifactorial biological process. It is recommended to revise this manuscript including relevant quantitative data, where available, to give a stronger support to the relative contribution (or relevance) of the different molecular and cellular mechanisms cited herein on neurodegeneration in the CA1 region of the hippocampus following ischemia. For example, concentrations or cellular levels of amyloid β and α-synuclein, values of gene and protein expression changes of key proteins (amyloid protein precursor, β-secretase, presenilin 1, presenilin 2 and tau) and relative extent of tau phosphorylation. In case that some or many of these data are not currently known it should be indicated, as these will focus unsettled relevant quantitative issues that deserve to be studied. Opening new perspectives or future research trends are relevant points in a review and should also contribute to increase citation scores.
A point missing in this review is that there is not any comment on the likely presence of different types of activated astrocytes in the CA1 region of the ischemic hippocampus or proximal regions at different times after ischemia. This latter point is of particular relevance because different types of activated astrocytes secrete different types of cytokines. Also, it has been shown that reactive astrocytes can contribute to Aβ production in Alzheimer’s disease, see [Blasko et al. (2000) Costimulatory effects of interferon-gamma and interleukin-1beta or tumor necrosis factor alpha on the synthesis of Abeta1-40 and Abeta1-42 by human astrocytes. Neurobiol Dis.7:682-689; Liang Y et al. (2020) Upregulation of Alzheimer's Disease Amyloid-β Protein Precursor in Astrocytes Both in vitro and in vivo. J. Alzheimers Dis. 76:1071-1082]. This should be improved in the revised manuscript.
Author Response
Review 2. -All text changes are in red and citation changes are highlighted in green. 1). This review addresses a relevant scientific issue, which is clearly presented. The authors have a long track of publications on this topic in good standing specialized journals. Although this review is well written, I have several concerns that should be addressed by the authors before I can recommend publication of this manuscript. My main concern is that the authors have recently published closely related reviews, which are cited as references in this work. Therefore, the authors should revise this manuscript to highlight the novel experimental facts and hypothesis that are not dealt with in their previous and recent reviews. Ad.1.The new sections of the manuscript include "Amyloid protein precursor ....... genes", "Amyloid staining", "Tau protein staining", "Alpha-synuclein staining", "Neuropathophysiology and neuropathology" and "Conclusions". 2). In addition, the analysis of the literature data looks very descriptive and the overall discussion is largely qualitative. This is a weakness in this work, since it is well known that ischemia-reperfusion injury is a complex and multifactorial biological process. It is recommended to revise this manuscript including relevant quantitative data, where available, to give a stronger support to the relative contribution (or relevance) of the different molecular and cellular mechanisms cited herein on neurodegeneration in the CA1 region of the hippocampus following ischemia. For example, concentrations or cellular levels of amyloid β and α-synuclein, values of gene and protein expression changes of key proteins (amyloid protein precursor, β-secretase, presenilin 1, presenilin 2 and tau) and relative extent of tau phosphorylation. In case that some or many of these data are not currently known it should be indicated, as these will focus unsettled relevant quantitative issues that deserve to be studied. Opening new perspectives or future research trends are relevant points in a review and should also contribute to increase citation scores. Ad. 2. Quantitative data on gene changes is now included in MS. Tau protein phosphorylation has been extended with new data available in the literature (see sections Tau protein stainig and Conclusions). Regarding the remaining molecules, there is a lack of quantitative data in the literature. Future research proposals are at the end of section Conclusions. 3). A point missing in this review is that there is not any comment on the likely presence of different types of activated astrocytes in the CA1 region of the ischemic hippocampus or proximal regions at different times after ischemia. This latter point is of particular relevance because different types of activated astrocytes secrete different types of cytokines. Also, it has been shown that reactive astrocytes can contribute to Aβ production in Alzheimer’s disease, see [Blasko et al. (2000) Costimulatory effects of interferon-gamma and interleukin-1beta or tumor necrosis factor alpha on the synthesis of Abeta1-40 and Abeta1-42 by human astrocytes. Neurobiol Dis.7:682-689; Liang Y et al. (2020) Upregulation of Alzheimer's Disease Amyloid-β Protein Precursor in Astrocytes Both in vitro and in vivo. J. Alzheimers Dis. 76:1071-1082]. This should be improved in the revised manuscript. Ad.3. The suggested literature and data from it have been added to MS on p. 7.
Round 2
Reviewer 2 Report
The authors have answered my questions/concerns raised to the first submission of this manuscript. Therefore, I recommend acceptance of this revised manuscript for publication with only the minor text corrections indicated below.
Minor text corrections:
Line 257. Please, revise whether it should read 1.371-fold change instead of 1,371-fold change (typo error?).
Line 392. I suggest to place a hyphen after amyloid to read: amyloid-dependent manner.
Line 402. There is not a reference 3147, please, correct this likely typo error. I must assume that a comma is lacking: 31,47 or 3,147?
Line 403. The expression “influenced the behavior of the tau protein” is rather unusual and inappropriate to refer to regulation/dysregulation of a protein function. Please, correct. For example, “altered the phosphorylation level (or extent) of tau protein”
Lines 435-439. The text of this long phrase needs to be improved for the shake of clarity, and also to minimize speculative assertions. Suggestion: “This review summarizes previous works suggesting that ischemia probably plays a significant role in triggering and/or accelerating the Alzheimer’s disease type cognitive decline. On these grounds, a better understanding of the molecular mechanisms of the progression of injury after hippocampus ischemia is still needed. Also, it is suggested to obtain more complete genetic, molecular, and cognitive profiles of patients to allow for future studies on an individual basis.”
Line 442. Please, change “this presentation” (this is not a lecture), for example, by “our current knowledge”.
Author Response
We have included all the last suggestions in the manuscript. Thanks.
